# NOT-SO-RANDOM FEATURES

**Brian Bullins**     **Cyril Zhang**     **Yi Zhang**
Department of Computer Science
Princeton University
Princeton, NJ 08544, USA
{bbullins, cyril.zhang, y.zhang}@cs.princeton.edu

## ABSTRACT

We propose a principled method for kernel learning, which relies on a Fourier-analytic characterization of translation-invariant or rotation-invariant kernels. Our method produces a sequence of feature maps, iteratively refining the SVM margin. We provide rigorous guarantees for optimality and generalization, interpreting our algorithm as online equilibrium-finding dynamics in a certain two-player min-max game. Evaluations on synthetic and real-world datasets demonstrate scalability and consistent improvements over related random features-based methods.

## 1 INTRODUCTION

Choosing the right kernel is a classic question that has riddled machine learning practitioners and theorists alike. Conventional wisdom instructs the user to select a kernel which captures the structure and geometric invariances in the data. Efforts to formulate this principle have inspired vibrant areas of study, going by names from *feature selection* to *multiple kernel learning* (MKL).

We present a new, principled approach for selecting a *translation-invariant* or *rotation-invariant* kernel to maximize the SVM classification margin. We first describe a kernel-alignment subroutine, which finds a peak in the Fourier transform of an *adversarially* chosen data-dependent measure. Then, we define an iterative procedure that produces a sequence of feature maps, progressively improving the margin. The resulting algorithm is strikingly simple and scalable.

Intriguingly, our analysis interprets the main algorithm as no-regret learning dynamics in a zero-sum min-max game, whose value is the classification margin. Thus, we are able to quantify convergence guarantees towards the largest margin realizable by a kernel with the assumed invariance. Finally, we exhibit experiments on synthetic and benchmark datasets, demonstrating consistent improvements over related random features-based kernel methods.

### 1.1 RELATED WORK

There is a vast literature on MKL, from which we use the key concept of *kernel alignment* (Cristianini et al., 2002). Otherwise, our work bears few similarities to traditional MKL; this and much related work (e.g. Cortes et al. (2012); Gönen & Alpaydın (2011); Lanckriet et al. (2004)) are concerned with selecting a kernel by combining a collection of base kernels, chosen beforehand. Our method allows for greater expressivity and even *better* generalization guarantees.

Instead, we take inspiration from the method of *random features* (Rahimi & Recht, 2007). In this pioneering work, originally motivated by scalability, feature maps are sampled according to the Fourier transform of a chosen kernel. The idea of optimizing a kernel in random feature space was studied by Sinha & Duchi (2016). In this work, which is most similar to ours, kernel alignment is optimized via importance sampling on a fixed, finitely supported proposal measure. However, the proposal can fail to contain informative features, especially in high dimension; indeed, they highlight efficiency, rather than showing performance improvements over RBF features.

Learning a kernel in the Fourier domain (without the primal feature maps) has also been considered previously: Oliva et al. (2016) and Yang et al. (2015b) model the Fourier spectrum parametrically, which limits expressivity; the former also require complicated posterior inference procedures.

Bǎzǎvan et al. (2012) study learning a kernel in the Fourier domain jointly with regression parameters. They show experimentally that this locates informative frequencies in the data, without theoretical guarantees. Our visualizations suggest this approach can get stuck in poor local minima, even in 2 dimensions.

Crammer et al. (2003) also use boosting to build a kernel sequentially; however, they only consider a basis of linear feature maps, and require costly generalized eigenvector computations. From a statistical view, Fukumizu et al. (2009) bound the SVM margin in terms of maximum mean discrepancy, which is equivalent to (unweighted) kernel alignment. Notably, their bound can be loose if the number of support vectors is small; in such situations, our theory provides a tighter characterization. Moreover, our attention to the margin goes beyond the usual objective of kernel alignment.

## 1.2 OUR CONTRIBUTIONS

We present an algorithm that outputs a sequence of Fourier features, converging to the maximum realizable SVM classification margin on a labeled dataset. At each iteration, a pair of features is produced, which maximizes kernel alignment with a changing, adversarially chosen measure. As this measure changes slowly, the algorithm builds a diverse and informative feature representation.

Our main theorem can be seen as a case of von Neumann's min-max theorem for a zero-sum concave-linear game; indeed, our method bears a deep connection to boosting (Freund & Schapire, 1996; Schapire, 1999). In particular, both the theory and empirical evidence suggest that the generalization error of our method *decreases* as the number of random features increases. In traditional MKL methods, generalization bounds worsen as base kernels are added.

Other methods in the framework of Fourier random features take the approach of approximating a kernel by sampling feature maps from a continuous distribution. In contrast, our method constructs a measure with small finite support, and realizes the kernel *exactly* by enumerating the associated finite-dimensional feature map; there is no randomness in the features.[1]

## 2 PRELIMINARIES

### 2.1 THE FOURIER DUAL MEASURE OF A KERNEL

We focus on two natural families of kernels $k(\mathbf{x}, \mathbf{x}')$: *translation-invariant* kernels on $\mathcal{X} = \mathbb{R}^d$, which depend only on $\mathbf{x} - \mathbf{x}'$, and *rotation-invariant* kernels on the hypersphere $\mathcal{X} = \mathbb{S}^{d-1}$, which depend only on $\langle \mathbf{x}, \mathbf{x}' \rangle$. These invariance assumptions subsume most widely-used classes of kernels; notably, the Gaussian (RBF) kernel satisfies both. For the former invariance, Bochner's theorem provides a Fourier-analytic characterization:

**Theorem 2.1** (e.g. Eq. (1), Sec. 1.4.3 in Rudin (2011)). *A translation-invariant continuous function $k : \mathbb{R}^d \times \mathbb{R}^d \to \mathbb{R}$ is positive semidefinite if and only if $k(\mathbf{x} - \mathbf{x}')$ is the Fourier transform of a symmetric non-negative measure (where the Fourier domain is $\Omega = \mathbb{R}^d$). That is,*

$$k(\mathbf{x} - \mathbf{x}') = \int_\Omega \lambda(\omega) \, e^{i\langle \omega, \mathbf{x} - \mathbf{x}' \rangle} \, d\omega = \int_\Omega \lambda(\omega) \, e^{i\langle \omega, \mathbf{x} \rangle} \, \overline{e^{i\langle \omega, \mathbf{x}' \rangle}} \, d\omega, \tag{1}$$

*for some $\lambda \in L_1(\Omega)$, satisfying $\lambda(\omega) \geq 0$ and $\lambda(\omega) = \lambda(-\omega)$ for all $\omega \in \Omega$.*

A similar characterization is available for rotation-invariant kernels, where the Fourier basis functions are the *spherical harmonics*, a countably infinite family of complex polynomials which form an orthonormal basis for square-integrable functions $\mathcal{X} \to \mathbb{C}$. To unify notation, let $\Omega \subset \mathbb{N} \times \mathbb{Z}$ be the set of valid index pairs $\omega = (\ell, m)$, and let $\omega \mapsto -\omega$ denote a certain involution on $\Omega$; we supply details and references in Appendix B.1.

**Theorem 2.2.** *A rotation-invariant continuous function $k : \mathbb{S}^{d-1} \times \mathbb{S}^{d-1} \to \mathbb{R}$ is positive semidefinite if and only if it has a symmetric non-negative expansion into spherical harmonics $Y^d_{\ell,m}$, i.e.*

$$k(\langle \mathbf{x}, \mathbf{x}' \rangle) = \sum_{j=0}^\infty \sum_{l=1}^{N(d,\ell)} \lambda(\ell, m) \, Y^d_{\ell,m}(\mathbf{x}) \, \overline{Y^d_{\ell,m}(\mathbf{x}')}, \tag{2}$$

---

[1]We note a superficial resemblance to quasi-Monte Carlo methods (Avron et al., 2016); however, these are concerned with accelerating convergence rates rather than learning a kernel.

*for some $\lambda \in L_1(\Omega)$, with $\lambda(\omega) \geq 0$ and $\lambda(\omega) = \lambda(-\omega)$ for all valid index pairs $\omega = (\ell, m) \in \Omega$.*

In each of these cases, we call this Fourier transform $\lambda_k(\omega)$ the *dual measure* of $k$. This measure decomposes $k$ into a non-negative combination of Fourier basis kernels. Furthermore, this decomposition gives us a *feature map* $\phi : \mathcal{X} \to L_2(\Omega, \lambda_k)$ whose image realizes the kernel under the codomain's inner product;[2] that is, for all $\mathbf{x}, \mathbf{x}' \in \mathcal{X}$,

$$k(\mathbf{x}, \mathbf{x}') = \int_\Omega \lambda_k(\omega) \, \phi_{\mathbf{x}}(\omega) \, \overline{\phi_{\mathbf{x}'}(\omega)} \, \mathrm{d}\omega. \tag{3}$$

Respectively, these feature maps are $\phi_{\mathbf{x}}(\omega) = e^{i\langle \omega, \mathbf{x} \rangle}$ and $\phi_{\mathbf{x}}(\ell, m) = Y_{\ell,m}^d(\mathbf{x})$. Although they are complex-valued, symmetry of $\lambda_k$ allows us to apply the transformation $\{\phi_{\mathbf{x}}(\omega), \phi_{\mathbf{x}}(-\omega)\} \mapsto \{\mathrm{Re}\,\phi_{\mathbf{x}}(\omega), \mathrm{Im}\,\phi_{\mathbf{x}}(\omega)\}$ to yield real features, preserving the inner product. The analogous result holds for spherical harmonics.

## 2.2 KERNEL ALIGNMENT

In a binary classification task with $n$ training samples ($\mathbf{x}_i \in \mathcal{X}, y_i \in \{\pm 1\}$), a widely-used quantity for measuring the quality of a kernel $k : \mathcal{X} \times \mathcal{X} \to \mathbb{R}$ is its *alignment* (Cristianini et al., 2002; Cortes et al., 2012),[3] defined by

$$\gamma_k(\mathbb{P}, \mathbb{Q}) \stackrel{\mathrm{def}}{=} \sum_{i,j \in [n]} k(\mathbf{x}_i, \mathbf{x}_j) y_i y_j = \mathbf{y}^T \mathbf{G}_k \mathbf{y},$$

where $\mathbf{y}$ is the vector of labels and $\mathbf{G}_k$ is the Gram matrix. Here, we let $\mathbb{P} = \sum_{i:y_i=1} \delta_{\mathbf{x}_i}$ and $\mathbb{Q} = \sum_{i:y_i=-1} \delta_{\mathbf{x}_i}$ denote the (unnormalized) empirical measures of each class, where $\delta_{\mathbf{x}}$ is the Dirac measure at $\mathbf{x}$. When $\mathbb{P}, \mathbb{Q}$ are arbitrary measures on $\mathcal{X}$, this definition generalizes to

$$\gamma_k(\mathbb{P}, \mathbb{Q}) \stackrel{\mathrm{def}}{=} \iint_{\mathcal{X}^2} k(\mathbf{x}, \mathbf{x}') \, \mathrm{d}\mathbb{P}^2(\mathbf{x}, \mathbf{x}') + \iint_{\mathcal{X}^2} k(\mathbf{x}, \mathbf{x}') \, \mathrm{d}\mathbb{Q}^2(\mathbf{x}, \mathbf{x}') - \iint_{\mathcal{X}^2} k(\mathbf{x}, \mathbf{x}') \, \mathrm{d}\mathbb{P}(\mathbf{x}) \, \mathrm{d}\mathbb{Q}(\mathbf{x}').$$

In terms of the dual measure $\lambda_k(\omega)$, kernel alignment takes a useful alternate form, noted by Sriperumbudur et al. (2010). Let $\mu$ denote the signed measure $\mathbb{P} - \mathbb{Q}$. Then, when $k$ is translation-invariant, we have

$$\gamma_k(\mathbb{P}, \mathbb{Q}) = \int_\Omega \lambda_k(\omega) \left| \int_{\mathcal{X}} e^{i\langle \omega, \mathbf{x} \rangle} \, \mathrm{d}\mu(\mathbf{x}) \right|^2 \mathrm{d}\omega \stackrel{\mathrm{def}}{=} \int_\Omega \lambda_k(\omega) \, v(\omega). \tag{4}$$

Analogously, when $k$ is rotation-invariant, we have

$$\gamma_k(\mathbb{P}, \mathbb{Q}) = \sum_{(\ell,m) \in \Omega} \lambda_k(\ell, m) \left| \int_{\mathcal{X}} Y_{\ell,m}^d(\mathbf{x}) \, \mathrm{d}\mu(\mathbf{x}) \right|^2 \stackrel{\mathrm{def}}{=} \sum_{(\ell,m) \in \Omega} \lambda_k(\ell, m) \, v(\ell, m). \tag{5}$$

It can also be verified that $\|\lambda_k\|_{L_1(\Omega)} = k(\mathbf{x}, \mathbf{x})$, which is of course the same for all $\mathbf{x} \in \mathcal{X}$. In each case, the alignment is *linear* in $\lambda_k$. We call $v(\omega)$ the *Fourier potential*, which is the squared magnitude of the Fourier transform of the signed measure $\mu = \mathbb{P} - \mathbb{Q}$. This function is clearly bounded pointwise by $(\mathbb{P}(\mathcal{X}) + \mathbb{Q}(\mathcal{X}))^2$.

## 3 ALGORITHMS

### 3.1 MAXIMIZING KERNEL ALIGNMENT IN THE FOURIER DOMAIN

First, we consider the problem of finding a kernel $k$ (subject to either invariance) that maximizes alignment $\gamma_k(\mathbb{P}, \mathbb{Q})$; we optimize the dual measure $\lambda_k(\omega)$. Aside from the non-negativity and symmetry constraints from Theorems 2.1 and 2.2, we constrain $\|\lambda_k\|_1 = 1$, as this quantity appears as a normalization constant in our generalization bounds (see Theorem 4.3). Maximizing $\gamma_k(\mathbb{P}, \mathbb{Q})$ in

---

[2]In fact, such a pair $(\phi, \lambda)$ exists for any psd kernel $k$; see Dai et al. (2014) or Devinatz (1953).

[3]Definitions vary up to constants and normalization, depending on the use case.

this constraint set, which we call $\mathcal{L} \subset L_1(\Omega)$, takes the form of a linear program on an infinite-dimensional simplex. Noting that $v(\omega) = v(-\omega) \geq 0$, $\gamma_k$ is maximized by placing a Dirac mass at any pair of opposite modes $\pm\omega^* \in \operatorname{argmax}_\omega v(\omega)$.

At first, $\mathbb{P}$ and $\mathbb{Q}$ will be the empirical distributions of the classes, specified in Section 2.2. However, as Algorithm 2 proceeds, it will reweight each data point $\mathbf{x}_i$ in the measures by $\boldsymbol{\alpha}(i)$. Explicitly, the *reweighted Fourier potential* takes the form[4]

$$v_{\boldsymbol{\alpha}}(\omega) \stackrel{\text{def}}{=} \left| \sum_{i=1}^n y_i \boldsymbol{\alpha}_i e^{\iota \langle \omega, \mathbf{x}_i \rangle} \right|^2 \quad \text{or} \quad v_{\boldsymbol{\alpha}}(\ell, m) \stackrel{\text{def}}{=} \left| \sum_{i=1}^n y_i \boldsymbol{\alpha}_i Y_{\ell,m}^d(\mathbf{x}_i) \right|^2.$$

Due to its non-convexity, maximizing $v_{\boldsymbol{\alpha}}(\omega)$, which can be interpreted as finding a global Fourier peak in the data, is theoretically challenging. However, we find that it is easy to find such peaks in our experiments, even in hundreds of dimensions. This arises from the empirical phenomenon that realistic data tend to be band-limited, a cornerstone hypothesis in data compression. An $\ell_2$ constraint (or equivalently, $\ell_2$ regularization; see Kakade et al. (2009)) can be explicitly enforced to promote band-limitedness; we find that this is not necessary in practice.

When a gradient is available (in the translation-invariant case), we use Langevin dynamics (Algorithm 1) to find the peaks of $v(\omega)$, which enjoys mild theoretical hitting-time guarantees (see Theorem 4.5). See Appendix A.3 for a discussion of the (discrete) rotation-invariant case.

---

**Algorithm 1** Langevin dynamics for kernel alignment

---

1: *Input:* training samples $S = \{(\mathbf{x}_i, y_i)\}_{i=1}^n$, weights $\boldsymbol{\alpha} \in \mathbb{R}^n$.
2: *Parameters:* time horizon $\tau$, diffusion rate $\zeta$, temperature $\xi$.
3: Initialize $\omega_0$ arbitrarily.
4: **for** $t = 0, \ldots, \tau - 1$ **do**
5:      Update $\omega_{t+1} \leftarrow \omega_t + \zeta \nabla v_{\boldsymbol{\alpha}}(\omega_t) + \sqrt{\frac{2\xi}{\zeta}} z$, where $z \sim \mathcal{N}(0, I)$.
6: **end for**
7: **return** $\omega^* := \operatorname{argmax}_t v(\omega_t)$

---

It is useful in practice to use parallel initialization, running $m$ concurrent copies of the diffusion process and returning the best single $\omega$ encountered. This admits a very efficient GPU implementation: the multi-point evaluations $v_{\boldsymbol{\alpha}}(\omega_{1..m})$ and $\nabla v_{\boldsymbol{\alpha}}(\omega_{1..m})$ can be computed from an $(m, d)$ by $(d, n)$ matrix product and pointwise trigonometric functions. We find that Algorithm 1 typically finds a reasonable peak within $\sim$100 steps.

### 3.2 Learning the Margin-Maximizing Kernel for SVM

Support vector machines (SVMs) are perhaps the most ubiquitous use case of kernels in practice. To this end, we propose a method that *boosts* Algorithm 1, building a kernel that maximizes the classification margin. Let $k$ be a kernel with dual $\lambda_k$, and $\mathbf{Y} \stackrel{\text{def}}{=} \operatorname{diag}(\mathbf{y})$. Write the dual $l_1$-SVM objective,[5] parameterizing the kernel by $\lambda_k$:

$$\begin{aligned} F(\boldsymbol{\alpha}, \lambda_k) &\stackrel{\text{def}}{=} \mathbf{1}^T \boldsymbol{\alpha} - \frac{1}{2} \boldsymbol{\alpha}^T \mathbf{Y} \mathbf{G}_k \mathbf{Y} \boldsymbol{\alpha} \\ &= \mathbf{1}^T \boldsymbol{\alpha} - \frac{1}{2} \gamma_k(\boldsymbol{\alpha}\mathbb{P}, \boldsymbol{\alpha}\mathbb{Q}) \quad = \mathbf{1}^T \boldsymbol{\alpha} - \frac{1}{2} \int_\Omega \lambda_k(\omega) \, v_{\boldsymbol{\alpha}}(\omega) \, d\omega. \end{aligned}$$

Thus, for a fixed $\boldsymbol{\alpha}$, $F$ is equivalent to kernel alignment, and can be minimized by Algorithm 1. However, the support vector weights $\boldsymbol{\alpha}$ are of course *not* fixed; given a kernel $k$, $\boldsymbol{\alpha}$ is chosen to maximize $F$, giving the (reciprocal) SVM margin. In all, to find a kernel $k$ which maximizes the margin under an *adversarial* choice of $\boldsymbol{\alpha}$, one must consider a two-player zero-sum game:

$$\min_{\lambda \in \mathcal{L}} \max_{\boldsymbol{\alpha} \in \mathcal{K}} F(\boldsymbol{\alpha}, \lambda), \tag{6}$$

---

[4]Whenever $i$ is an index, we will denote the imaginary unit by $\iota$.
[5]Of course, our method applies to $l_2$ SVMs, and has even stronger theoretical guarantees; see Section 4.1.

where $\mathcal{L}$ is the same constraint set as in Section 3.1, and $\mathcal{K}$ is the usual dual feasible set $\{0 \preccurlyeq \boldsymbol{\alpha} \preccurlyeq C, \mathbf{y}^T \boldsymbol{\alpha} = 0\}$ with box parameter $C$.

In this view, we make some key observations. First, Algorithm 1 allows the min-player to play a pure-strategy *best response* to the max-player. Furthermore, a mixed strategy $\bar{\lambda}$ for the min-player is simply a translation- or rotation-invariant kernel, realized by the feature map corresponding to its support. Finally, since the objective is linear in $\lambda$ and concave in $\boldsymbol{\alpha}$, there exists a Nash equilibrium $(\lambda^*, \boldsymbol{\alpha}^*)$ for this game, from which $\lambda^*$ gives the margin-maximizing kernel.

We can use no-regret learning dynamics to approximate this equilibrium. Algorithm 2 runs Algorithm 1 for the min-player, and online gradient ascent (Zinkevich, 2003) for the max-player. Intuitively (and as is visualized in our synthetic experiments), this process slowly morphs the landscape of $v_{\boldsymbol{\alpha}}$ to emphasize the margin, causing Algorithm 1 to find progressively more informative features. At the end, we simply concatenate these features; contingent on the success of the kernel alignment steps, we have approximated the Nash equilibrium.

---

**Algorithm 2** No-regret learning dynamics for SVM margin maximization

---

1: *Input:* training samples $S = \{(\mathbf{x}_i, y_i)\}_{i=1}^n$.
2: *Parameters:* box constraint $C$, # steps $T$, step sizes $\{\eta_t\}$; parameters for Algorithm 1.
3: Set $\phi_{\mathbf{x}}(\omega) = e^{i\langle \omega, \mathbf{x} \rangle}$ *(translation-invariant)*, or $\phi_{\mathbf{x}}(\ell, m) = Y_{\ell,m}^d(\mathbf{x})$ *(rotation-invariant)*.
4: Initialize $\boldsymbol{\alpha} = \text{Proj}_{\mathcal{K}}[\frac{C}{2} \cdot \mathbf{1}]$.
5: **for** $t = 1, \ldots, T$ **do**
6:     Use Algorithm 1 (or other $v_{\boldsymbol{\alpha}}$ maximizer) on $S$ with weights $\boldsymbol{\alpha}_t$, returning $\omega_t$.
7:     Append two features $\{\text{Re}\,\phi_{\mathbf{x}'}(\omega_t), \text{Im}\,\phi_{\mathbf{x}'}(-\omega_t)\}$ to each $\mathbf{x}_i$'s representation $\Phi(\mathbf{x}_i)$.
8:     Compute gradient $\mathbf{g}_t := \nabla_{\boldsymbol{\alpha}} F(\boldsymbol{\alpha}_t, \lambda_t)$, where $\lambda_t = \delta_{\omega_t} + \delta_{-\omega_t}$.
9:     Update $\boldsymbol{\alpha}_{t+1} \leftarrow \text{Proj}_{\mathcal{K}}[\boldsymbol{\alpha}_t + \eta_t \mathbf{g}_t]$.
10: **end for**
11: **return** features $\{\Phi(\mathbf{x}_i) \in \mathbb{R}^{2T}\}_{i=1}^n$, or dual measure $\bar{\lambda} := \frac{1}{2T} \sum_{t=1}^T \delta_{\omega_t} + \delta_{-\omega_t}$.

---

We provide a theoretical analysis in Section 4.1, and detailed discussion on heuristics, hyperparameters, and implementation details in depth in Appendix A. One important note is that the online gradient $g_t = \mathbf{1} - 2\mathbf{Y}\,\text{Re}(\langle \Phi_t, \mathbf{Y}\boldsymbol{\alpha}_t \rangle \overline{\Phi_t})$ is computed very efficiently, where $\Phi_t(i) = \phi_{\mathbf{x}_i}(\omega_t) \in \mathbb{C}^m$ is the vector of the most recently appended features. Langevin dynamics, easily implemented on a GPU, comprise the primary time bottleneck.

## 4 THEORY

### 4.1 CONVERGENCE OF NO-REGRET LEARNING DYNAMICS

We first state the main theoretical result, which quantifies the convergence properties of Algorithm 2.

**Theorem 4.1** (Main). *Assume that at each step $t$, Algorithm 1 returns an $\varepsilon_t$-approximate global maximizer $\omega_t$ (i.e., $v_{\boldsymbol{\alpha}_t}(\omega_t) \geq \sup_{\omega \in \Omega} v_{\boldsymbol{\alpha}_t}(\omega) - \varepsilon_t$). Then, with a certain choice of step sizes $\eta_t$, Algorithm 2 produces a dual measure $\bar{\lambda} \in \mathcal{L}$ which satisfies*

$$\max_{\boldsymbol{\alpha} \in \mathcal{K}} F(\boldsymbol{\alpha}, \bar{\lambda}) \leq \min_{\lambda \in \mathcal{L}} \max_{\boldsymbol{\alpha} \in \mathcal{K}} F(\boldsymbol{\alpha}, \lambda) + \frac{\sum_{t=1}^T \varepsilon_t}{2T} + \frac{3(C + 2C^2)n}{4\sqrt{T}}.$$

(Alternate form.) *Suppose instead that $v_{\boldsymbol{\alpha}_t}(\omega_t) \geq \rho$ at each time $t$. Then, $\bar{\lambda}$ satisfies*

$$\max_{\boldsymbol{\alpha} \in \mathcal{K}} F(\boldsymbol{\alpha}, \bar{\lambda}) \leq \rho + \frac{3(C + 2C^2)n}{4\sqrt{T}}.$$

We prove Theorem 4.1 in Appendix C. For convenience, we state the first version as an explicit margin bound (in terms of competitive ratio $M/M^*$):

**Corollary 4.2.** *Let $M$ be the margin obtained by training an $\ell_1$ linear SVM with the same $C$ as in Algorithm 2, on the transformed samples $\{(\Phi(\mathbf{x}_i), y_i)\}$. Then, $M$ is $(1 - \delta)$-competitive with $M^*$, the maximally achievable margin by a kernel with the assumed invariance, with*

$$\delta \leq \frac{\sum_t \varepsilon_t}{T} + \frac{3(C + 2C^2)n}{4\sqrt{T}}.$$

This bound arises from the regret analysis of online gradient ascent (Zinkevich, 2003); our analysis is similar to the approach of Freund & Schapire (1996), where they present a boosting perspective. When using an $\ell_2$-SVM, the final term can be improved to $O(\frac{\log T}{T})$ (Hazan et al., 2007). For a general overview of results in the field, refer to Hazan (2016).

## 4.2 GENERALIZATION BOUNDS

Finally, we state two (rather distinct) generalization guarantees. Both depend mildly on a *bandwidth* assumption $\|\omega\|_2 \leq R_\omega$ and the norm of the data $R_x \stackrel{\text{def}}{=} \max_i \|\mathbf{x}_i\|$. First, we state a margin-dependent SVM generalization bound, due to Koltchinskii & Panchenko (2002). Notice the appearance of $\|\lambda_k\|_1 = R_\lambda$, justifying our choice of normalization constant for Algorithm 1. Intriguingly, the end-to-end generalization error of our method *decreases* with an increasing number of random features, since the margin bound is being refined during Algorithm 2.

**Theorem 4.3** (Generalization via margin). *For any SVM decision function $f : \mathcal{X} \to \mathbb{R}$ with a kernel $k_\lambda$ constrained by $\|\lambda\|_1 \leq R_\lambda$ trained on samples $S$ drawn i.i.d. from distribution $\mathcal{D}$, the generalization error is bounded by*

$$\Pr_{(\mathbf{x},y)\sim\mathcal{D}}[yf(\mathbf{x}) \leq 0] \leq \min_\theta \frac{1}{n}\sum_{i=1}^n \mathbf{1}_{y_i f(x_i) \leq \theta} + \frac{6R_\omega R_x}{\theta}\sqrt{\frac{R_\lambda}{n}} + 3\sqrt{\frac{\log\frac{2}{\delta}}{n}}.$$

The proof can be found in Appendix D. Note that this improves on the generic result for MKL, from Theorem 2 in (Cortes et al., 2010), which has a $\sqrt{\log T}$ dependence on the number of base kernels $T$. This improvement stems from the rank-one property of each component kernel.

Next, we address another concern entirely: the sample size required for $v(\omega)$ to approximate the ideal Fourier potential $v_{\text{ideal}}(\omega)$, the squared magnitude of the Fourier transform of the signed measure $\mathbb{P} - \mathbb{Q}$ arising from the true distribution. For the shift-invariant case:

**Theorem 4.4** (Generalization of the potential). *Let $S$ be a set of i.i.d. training samples on $\mathcal{D}$, with $n \geq O\left(\frac{d\log\frac{1}{\varepsilon}+\log\frac{1}{\delta}}{\varepsilon^2}\right)$. We have that for all $\omega : \|\omega\| \leq R_\omega$, with probability at least $1 - \delta$,*

$$\left|\frac{v(\omega)}{n^2} - v_{\text{ideal}}(\omega)\right| \leq \varepsilon.$$

*The $O(\cdot)$ suppresses factors polynomial in $R_x$ and $R_\omega$.*

The full statement and proof are standard, and deferred to Appendix E. In particular, this result allows for a mild guarantee of polynomial hitting-time on the locus of approximate local maxima of $v_{\text{ideal}}$ (as opposed to the empirical $v$). Adapting the main result from Zhang et al. (2017):

**Theorem 4.5** (Langevin hitting time). *Let $\omega_\tau$ be the output of Algorithm 1 on $v_{\boldsymbol{\alpha}}(\omega)$, after $\tau$ steps. Algorithm 1 finds an approximate local maximum of $v_{\text{ideal}}$ in polynomial time. That is, with $U$ being the set of $\varepsilon$-approximate local maxima of $v_{\text{ideal}}(\omega)$, some $\omega_t$ satisfies*

$$v(\omega_t) \geq \inf_{d(U,\omega)\leq\Delta} v(\omega)$$

*for some $\tau \leq O(\text{poly}\left(R_x, R_\omega, d, \zeta, \xi, \varepsilon^{-1}, \Delta^{-1}, \log(1/\delta)\right)$, with probability at least $1 - \delta$.*

Of course, one should not expect polynomial hitting time on approximate *global* maxima; Roberts & Tweedie (1996) give asymptotic mixing guarantees.

## 5 EXPERIMENTS

In this section, we highlight the most important and illustrative parts of our experimental results. For further details, we provide an extended addendum to the experimental section in Appendix A. The code can be found at `github.com/yz-ignescent/Not-So-Random-Features`.

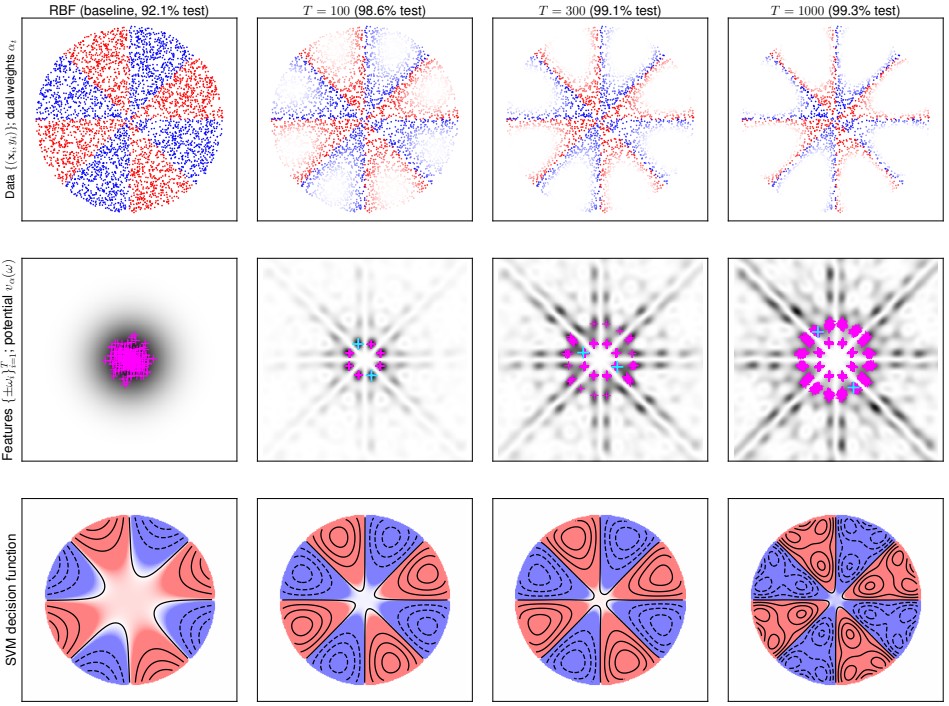

(a) Evolution of Algorithm 2. *Top:* The windmill dataset, weighted by dual variables $\boldsymbol{\alpha}_t$. *Middle:* $2t$ random features (magenta; last added $\pm\omega_t$ in cyan), overlaying the Fourier potential $v_{\boldsymbol{\alpha}_t}(\omega)$ in the background. *Bottom:* Decision boundary of a hinge-loss classifier trained on the kernel, showing refinement at the margin. Contours indicate the value of the margin.

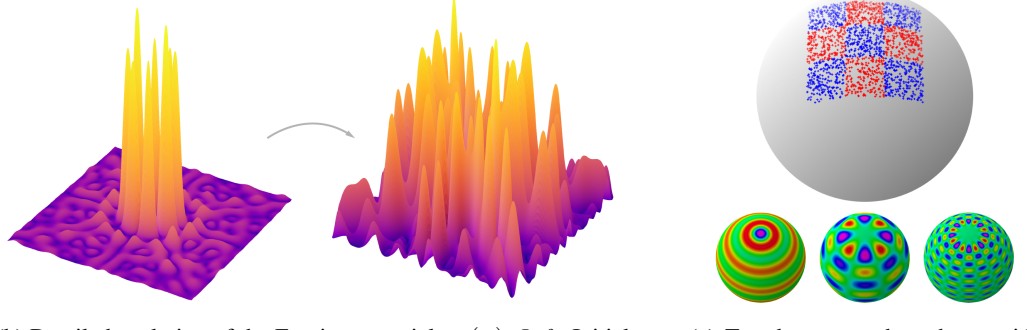

(b) Detailed evolution of the Fourier potential $v_{\boldsymbol{\alpha}}(\omega)$. *Left:* Initial potential $v(\omega)$, with uniform weights. *Right:* Reweighted $v_{\boldsymbol{\alpha}}(\omega)$ at $t = 300$. Note the larger support of peaks.

(c) Toy dataset on the sphere, with top-3 selected features: spherical harmonics $Y_{0,11}, Y_{4,9}, Y_{8,27}$.

Figure 1: **Visualizations and experiments on synthetic data.**

First, we exhibit two simple binary classification tasks, one in $\mathbb{R}^2$ and the other on $\mathbb{S}^2$, to demonstrate the power of our kernel selection method. As depicted in Figure 1, we create datasets with sharp boundaries, which are difficult for the standard RBF and *arccosine* (Cho & Saul, 2009) kernel. In both cases, $n_{\text{train}} = 2000$ and $n_{\text{test}} = 50000$.[6] Used on a $\ell_1$-SVM classifier, these baseline kernels saturate at 92.1% and 95.1% test accuracy, respectively; they are not expressive enough.

On the $\mathbb{R}^2$ "windmill" task, Algorithm 2 chooses random features that progressively refine the decision boundary at the margin. By $T = 1000$, it exhibits almost perfect classification (99.7% training, 99.3% test). Similarly, on the $\mathbb{S}^2$ "checkerboard" task, Algorithm 2 (with some adaptations described

---

[6]$n_{\text{test}}$ is chosen so large to measure the true generalization error.

in Appendix A.3) reaches almost perfect classification (99.7% training, 99.1% test) at $T = 100$, supported on only 29 spherical harmonics as features.

We provide some illuminating visualizations. Figures 1a and 1b show the evolution of the dual weights, random features, and classifier. As the theory suggests, the objective evolves to assign higher weight to points near the margin, and successive features improve the classifier's decisiveness in challenging regions (1a, bottom). Figure 1c visualizes some features from the $\mathbb{S}^2$ experiment.

Next, we evaluate our kernel on standard benchmark binary classification tasks. Challenging label pairs are chosen from the MNIST (LeCun et al., 1998) and CIFAR-10 (Krizhevsky, 2009) datasets; each task consists of $\sim$10000 training and $\sim$2000 test examples; this is considered to be large-scale for kernel methods. Following the standard protocol from Yu et al. (2016), 512-dimensional HoG features (Dalal & Triggs, 2005) are used for the CIFAR-10 tasks instead of raw images. Of course, our intent is not to show state-of-the-art results on these tasks, on which deep neural networks easily dominate. Instead, the aim is to demonstrate viability as a scalable, principled kernel method.

We compare our results to baseline random features-based kernel machines: the standard RBF random features (RBF-RF for short), and the method of Sinha & Duchi (2016) (LKRF),[7] using the same $\ell_1$-SVM throughout. As shown by Table 1, our method reliably outperforms these baselines, most significantly in the regime of few features. Intuitively, this lines up with the expectation that isotropic random sampling becomes exponentially unlikely to hit a good peak in high dimension; our method *searches* for these peaks.

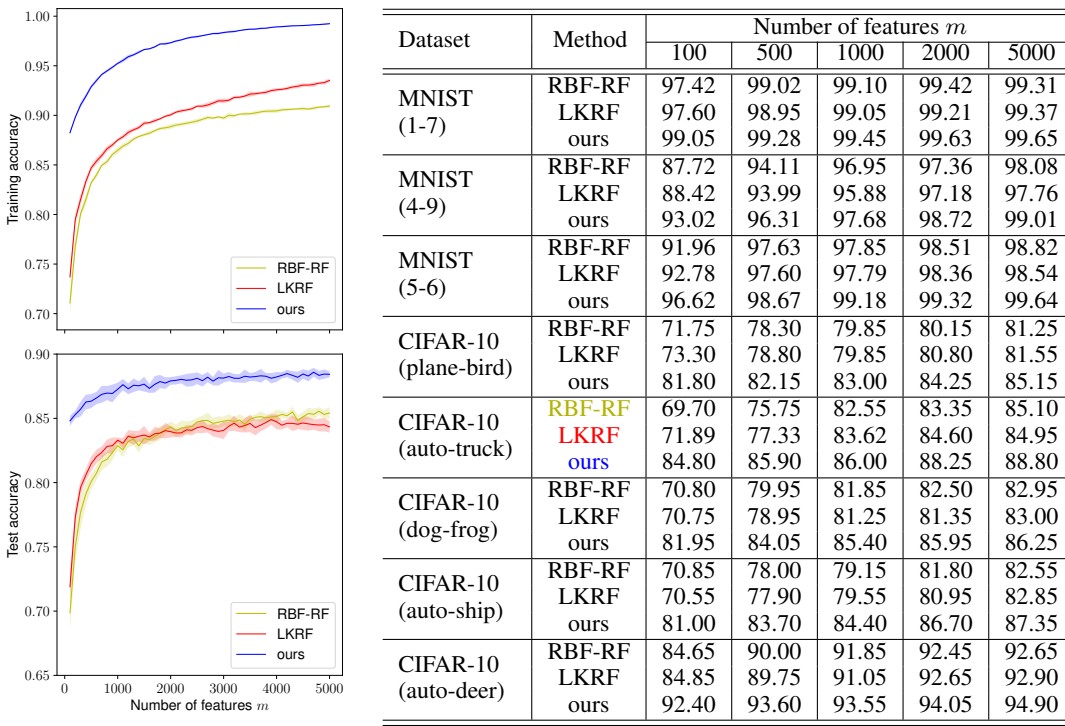

| Dataset | Method | Number of features $m$ | | | | |
|---|---|---|---|---|---|---|
| | | 100 | 500 | 1000 | 2000 | 5000 |
| MNIST (1-7) | RBF-RF | 97.42 | 99.02 | 99.10 | 99.42 | 99.31 |
| | LKRF | 97.60 | 98.95 | 99.05 | 99.21 | 99.37 |
| | ours | 99.05 | 99.28 | 99.45 | 99.63 | 99.65 |
| MNIST (4-9) | RBF-RF | 87.72 | 94.11 | 96.95 | 97.36 | 98.08 |
| | LKRF | 88.42 | 93.99 | 95.88 | 97.18 | 97.76 |
| | ours | 93.02 | 96.31 | 97.68 | 98.72 | 99.01 |
| MNIST (5-6) | RBF-RF | 91.96 | 97.63 | 97.85 | 98.51 | 98.82 |
| | LKRF | 92.78 | 97.60 | 97.79 | 98.36 | 98.54 |
| | ours | 96.62 | 98.67 | 99.18 | 99.32 | 99.64 |
| CIFAR-10 (plane-bird) | RBF-RF | 71.75 | 78.30 | 79.85 | 80.15 | 81.25 |
| | LKRF | 73.30 | 78.80 | 79.85 | 80.80 | 81.55 |
| | ours | 81.80 | 82.15 | 83.00 | 84.25 | 85.15 |
| CIFAR-10 (auto-truck) | RBF-RF | 69.70 | 75.75 | 82.55 | 83.35 | 85.10 |
| | LKRF | 71.89 | 77.33 | 83.62 | 84.60 | 84.95 |
| | ours | 84.80 | 85.90 | 86.00 | 88.25 | 88.80 |
| CIFAR-10 (dog-frog) | RBF-RF | 70.80 | 79.95 | 81.85 | 82.50 | 82.95 |
| | LKRF | 70.75 | 78.95 | 81.25 | 81.35 | 83.00 |
| | ours | 81.95 | 84.05 | 85.40 | 85.95 | 86.25 |
| CIFAR-10 (auto-ship) | RBF-RF | 70.85 | 78.00 | 79.15 | 81.80 | 82.55 |
| | LKRF | 70.55 | 77.90 | 79.55 | 80.95 | 82.85 |
| | ours | 81.00 | 83.70 | 84.40 | 86.70 | 87.35 |
| CIFAR-10 (auto-deer) | RBF-RF | 84.65 | 90.00 | 91.85 | 92.45 | 92.65 |
| | LKRF | 84.85 | 89.75 | 91.05 | 92.65 | 92.90 |
| | ours | 92.40 | 93.60 | 93.55 | 94.05 | 94.90 |

Table 1: **Comparison on binary classification tasks.** Our method is compared against standard RBF random features (RBF-RF), as well as the method from Sinha & Duchi (2016) (LKRF). *Left:* Performance is measured on the CIFAR-10 automobile vs. truck task, varying the number of features $m$. Standard deviations over 10 trials are shown, demonstrating high stability.

Furthermore, our theory predicts that the margin keeps improving, regardless of the dimensionality of the feature maps. Indeed, as our classifier saturates on the training data, test accuracy continues increasing, without overfitting. This decoupling of generalization from model complexity is characteristic of boosting methods.

---

[7]Traditional MKL methods are not tested here, as they are noticeably ($> 100$ times) slower.

In practice, our method is robust with respect to hyperparameter settings. As well, to outperform both RBF-RF and LKRF with 5000 features, our method only needs $\sim 100$ features. Our GPU implementation reaches this point in $\sim 30$ seconds. See Appendix A.1 for more tuning guidelines.

## 6 CONCLUSION

We have presented an efficient kernel learning method that uses tools from Fourier analysis and online learning to optimize over two natural infinite families of kernels. With this method, we show meaningful improvements on benchmark tasks, compared to related random features-based methods. Many theoretical questions remain, such as accelerating the search for Fourier peaks (e.g. Hassanieh et al. (2012); Kapralov (2016)). These, in addition to applying our learned kernels to state-of-the-art methods (e.g. convolutional kernel networks (Mairal et al., 2014; Yang et al., 2015a; Mairal, 2016)), prove to be exciting directions for future work.

## ACKNOWLEDGMENTS

We are grateful to Sanjeev Arora, Roi Livni, Pravesh Kothari, Holden Lee, Karan Singh, and Naman Agarwal for helpful discussions. This research was supported by the National Science Foundation (NSF), the Office of Naval Research (ONR), and the Simons Foundation. The first two authors are supported in part by Elad Hazan's NSF grant IIS-1523815.

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

# A  APPENDIX FOR EXPERIMENTS

Algorithm 2 gives a high-level outline of the essential components of our method. However, it conceals several hyperparameter choices and algorithmic heuristics, which are pertinent when applying our method in practice. We discuss a few more details in this section.

## A.1  HYPERPARAMETER TUNING GUIDE

Throughout all experiments presented, we use hinge-loss SVM classifiers with $C = 1$. Note that the convergence of Algorithm 2 depends quadratically on $C$.

With regard to Langevin diffusion (Algorithm 1), we observe that the best samples arise from using high temperatures and Gaussian parallel initialization. For the latter, a rule-of-thumb is to initialize 500 parallel copies of Langevin dynamics, drawing the initial position $\{\omega_0\}$ from a centered isotropic Gaussian with $1.5\times$ the variance of the optimal RBF random features. (In turn, a common rule-of-thumb for this bandwidth is the median of pairwise Euclidean distances between data points.)

The step size in Algorithm 1 is tuned based on the magnitude of the gradient on $v(\omega)$, which can be significantly smaller than the upper bound derived in Section E. As is standard practice in Langevin Monte Carlo methods, the temperature is chosen so that the pertubation is roughly at the same magnitude as the gradient step. Empirically, running Langevin dynamics for $\sim 100$ steps suffices to locate a reasonably good peak. To further improve efficiency, one can modify Algorithm 1 to pick the top $k \approx 10$ samples, a $k$-fold speedup which does not degrade the features much.

The step size of online gradient ascent is set to balance between being conservative and promoting diverse samples; these steps should not saturate (thereby solving the dual SVM problem), in order to have the strongest regret bound. In our experiments, we find that the step size achieving the standard regret guarantee (scaling as $1/\sqrt{T}$) tends to be a little too conservative.

On the other hand, it *never* hurts (and seems important in practice) to saturate the peak-finding routine (Algorithm 1), since this contributes an additive improvement to the margin bound. Noting that the objective is very smooth (the $k$-th derivative scales as $R_x^k$), it may be beneficial to refine the samples using a few steps of gradient descent with a very small learning rate, or an accelerated algorithm for finding approximate local minima of smooth non-convex functions; see, e.g. Agarwal et al. (2017).

## A.2  PROJECTION ONTO THE SVM DUAL FEASIBLE SET

A quick note on projection onto the feasible set $\mathcal{K} = \{0 \preccurlyeq \boldsymbol{\alpha} \preccurlyeq C, \mathbf{y}^T \boldsymbol{\alpha} = 0\}$ of the SVM dual convex program: it typically suffices in practice to use alternating projection. This feasible set is the intersection of a hyperplane and a hypercube; both of which admit a simple projection step. The alternation projection onto the intersection of two non-empty convex sets was originally proposed by von Neumann (1949). The convergence rate can be shown to be linear. To obtain the dual variables in our experiments, we use 10 such alternating projections. This results in a dual feasible solution up to hardware precision, and is a negligible component of the total running time (for which the parallel gradient computations are the bottleneck).

---

**Algorithm 3** Alternating Projection for SVM Dual Constraints

---

 *Input:* $\boldsymbol{\alpha} \in \mathbb{R}^n$
 *Parameters:* box constraint $C$, label vector $\mathbf{y}$.
 **repeat**
  Project onto the box: $\boldsymbol{\alpha} = \mathsf{clip}(\boldsymbol{\alpha}, 0, C)$
  Project onto the hyperplane: $\boldsymbol{\alpha} \leftarrow \boldsymbol{\alpha} - \frac{\mathbf{y}^T \boldsymbol{\alpha}}{n} \mathbf{y}$
 **until** convergence
 **return** $\boldsymbol{\alpha}$

---

### A.3 SAMPLING SPHERICAL HARMONICS

As we note in Section 3.1, it is unclear how to define gradient Langevin dynamics on $v(l,m)$ in the inner-product case, since no topology is available on the indices $(l,m)$ of the spherical harmonics. One option is to emulate Langevin dynamics, by constructing a discrete Markov chain which mixes to $\lambda(l,m) \propto e^{\beta v(l,m)}$.

However, we find in our experiments that it suffices to compute $\lambda(l,m)$ by examining *all* values of $v(l,m)$ with $j$ no more than some threshold $J$. One should view this as approximating the kernel-target alignment objective function via Fourier truncation. This is highly parallelizable: it involves approximately $N(m,d)$ degree-$J$ polynomial evaluations on the same sample data, which can be expressed using matrix multiplication. In our experiments, it sufficed to examine the first 1000 coefficients; we remark that it is unnatural in any real-world datasets to expect that $v(\omega)$ only has large values outside the threshold $J$.

Under Fourier truncation, the domain of $\lambda$ becomes a finite-dimensional simplex. In the game-theoretic view of 4.1, an approximate Nash equilibrium becomes concretely achievable via Nesterov's excessive gap technique (Nesterov, 2005; Daskalakis et al., 2015), given that the kernel player's actions are restricted to a mixed strategy over a finite set of basis kernels.

Finally, we note a significant advantage to this setting, where we have a discrete set of Fourier coefficients: the same feature might be found multiple times. When a duplicate feature is found, it need not be concatenated to the representation; instead, the existing feature is scaled appropriately. This accounts for the drastically smaller support of features required to achieve near-perfect classification accuracy.

## B MORE ON SPHERICAL HARMONICS

In this section, we go into more detail about the spherical harmonics in $d$ dimensions. Although all of this material is standard in harmonic analysis, we provide this section for convenience, isolating only the relevant facts.

### B.1 SPHERICAL HARMONIC EXPANSION OF A ROTATION-INVARIANT KERNEL

First, we provide a proof sketch for Theorem 2.2. We rely on the following theorem, an analogue of Bochner's theorem on the sphere, which characterizes rotation-invariant kernels:

**Theorem B.1** (Schoenberg (1942)). *A continuous rotation-invariant function $k(\mathbf{x}, \mathbf{x}') = k(\langle \mathbf{x}, \mathbf{x}' \rangle)$ on $\mathbb{S}^{d-1} \times \mathbb{S}^{d-1}$ is positive semi-definite if and only if its expansion into Gegenbauer polynomials $P_i^d$ has only non-negative coefficients, i.e.*

$$k(\langle \mathbf{x}, \mathbf{x}' \rangle) = \sum_{m=0}^{\infty} \lambda_m P_m^d(\langle \mathbf{x}, \mathbf{x}' \rangle), \tag{7}$$

*with $\lambda_m \geq 0$, $\forall m \in \mathbb{N}^+$.*

The Gegenbauer polynomials $P_m^d : [-1, 1] \to \mathbb{R}$ are a generalization of the Legendre and Chebyshev polynomials; they form an orthogonal basis on $[-1, 1]$ under a certain weighting function (see Stein & Weiss (2016); Gallier (2009); Müller (2012) for details). Note that we adopt a different notation from Schoenberg (1942), which denotes our $P_m^d$ by $P_m^{(d-1)/2}$.

Unlike Theorem 2.1, Theorem B.1 alone does not provide a clear path for constructing a feature map for inner-product kernel, because the inputs $\mathbf{x}, \mathbf{x}'$ of the kernel are still coupled after the expansion into $P_m^d$, which acts on the inner product $\langle \mathbf{x}, \mathbf{x}' \rangle$ instead of on $\mathbf{x}$ and $\mathbf{x}'$ separately. Fortunately, (see, e.g., Proposition 1.18 in Gallier (2009)), the Gegenbauer polynomials evaluated on an inner product admits a decomposition into spherical harmonics $Y_{\ell,m}^d$:

$$P_m^d(\langle \mathbf{x}, \mathbf{x}' \rangle) = \frac{|\mathbb{S}^{d-1}|}{N(d,\ell)} \sum_{l=1}^{N(d,\ell)} Y_{\ell,m}^d(\mathbf{x}) \overline{Y_{\ell,m}^d(\mathbf{x}')} \quad \forall \mathbf{x}, \mathbf{x}' \in \mathbb{S}^{d-1}, \tag{8}$$

where $|\mathbb{S}^{d-1}|$ denotes the surface area of $\mathbb{S}^{d-1}$, and $N(d, \ell) = \binom{d-1+\ell}{\ell} - \binom{d-1+\ell}{\ell-2}$. Here, we use the normalization convention that $\int_{S^{d-1}} |Y_{\ell,m}^d(\mathbf{x})|^2 \, d\mathbf{x} = 1$. From the existence of this expansion follows the claim from Theorem 2.2. For a detailed reference, see Müller (2012).

### B.2 Pairing the Spherical Harmonics

We now specify the involution $\omega \mapsto -\omega$ on indices of spherical harmonics, which gives a pairing that takes the role of opposite Fourier coefficients in the $\mathcal{X} = \mathbb{R}^d$ case. In particular, the Fourier transform $\lambda$ of a real-valued function $k : \mathbb{R}^n \to \mathbb{R}$ satisfies $\lambda(\omega) = \lambda(-\omega)$, so that the dual measure can be constrained to be symmetric.

Now, consider the $\mathcal{X} = \mathbb{S}^{d-1}$ case, where the Fourier coefficients are on the set $\Omega$ of valid indices of spherical harmonics. We would like a permutation on the indices $\sigma$ so that $\sigma(\sigma(\omega)) = \omega$, and $\lambda(\omega) = \lambda(\sigma(\omega))$ whenever $\lambda$ is the spherical harmonic expansion of a real kernel.

For a fixed dimension $d$ and degree $\ell$, the spherical harmonics form an $N := N(d, \ell)$-dimensional orthogonal basis for a vector space $V$ of complex polynomials from $\mathbb{S}^{d-1}$ to $\mathbb{C}$, namely the $\ell$-homogeneous polynomials (with domain extended to $\mathbb{R}^n$) which are harmonic (eigenfunctions of the Laplace-Beltrami operator $\Delta_{\mathbb{S}^{d-1}}$). This basis is not unique; an arbitrary choice of basis may not contain pairs of conjugate functions.

Fortunately, such a basis exists constructively, by separating the Laplace-Beltrami operator over spherical coordinates $(\theta_1, \ldots, \theta_{d-1})$. Concretely, for a fixed $d$ and $\ell$, the $d$-dimensional spherical harmonic, indexed by $|\ell_1| \le \ell_2 \le \ell_3 \le \ldots \le \ell_{d-1}$, is defined as:

$$Y_{\ell_1, \ldots, \ell_N} \overset{\text{def}}{=} \frac{1}{\sqrt{2\pi}} e^{i\ell_1 \theta_1} \prod_{k=2}^{d-1} {}_k \overline{P}_{\ell_k}^{\ell_{d-1}}(\theta_k),$$

where the functions in the product come from a certain family of associated Legendre functions in $\sin \theta$. For a detailed treatment, we adopt the construction and conventions from Higuchi (1987).

In the scope of this paper, the relevant fact is that the desired involution is well-defined in all dimensions. Namely, consider the permutation $\sigma$ that sends a spherical harmonic indexed by $\omega = (d, \ell, \ell_1, \ell_2, \ldots, \ell_{d-1})$ to $\sigma(\omega) = (d, \ell, -\ell_1, \ell_2, \ldots, \ell_{d-1})$. Then, $Y_{-\omega}(\mathbf{x}) \overset{\text{def}}{=} Y_{\sigma(\omega)}(\mathbf{x}) = \overline{Y_\omega(\mathbf{x})}$ for all $\omega, \mathbf{x}$.

The symmetry condition in Theorem 2.2 follows straightforwardly. By orthonormality, we know that every square-integrable function $f : \mathbb{R}^n \to \mathbb{C}$ has a unique decomposition into spherical harmonics, with coefficients $\lambda_\omega = \langle f, Y_\omega \rangle$, so that $\lambda_{-\omega} = \langle f, \overline{Y_\omega} \rangle = \overline{\lambda_\omega}$. When $f$ is real-valued, we conclude that $\lambda_{-\omega} = -\lambda_\omega$, as claimed.

## C  Proof of the Main Theorem

In this section, we prove the main theorem, which quantifies convergence of Algorithm 2 to the Nash equilibrium. We restate it here:

**Theorem 4.1.** Assume that during each timestep $t$, the call to Algorithm 1 returns an $\varepsilon_t$-approximate global maximizer $\omega_t$ (i.e. $\hat{v}_{\boldsymbol{\alpha}_t}(\omega_t) \ge \max_{\omega \in \mathcal{K}} \hat{v}_{\boldsymbol{\alpha}_t}(\omega) - \varepsilon_t$). Then, Algorithm 2 returns a dual measure $\bar{\lambda}$, which satisfies

$$\max_{\boldsymbol{\alpha} \in \mathcal{K}} F(\boldsymbol{\alpha}, \bar{\lambda}) \le \min_{\lambda \in \mathcal{L}} \max_{\boldsymbol{\alpha} \in \mathcal{K}} F(\boldsymbol{\alpha}, \lambda) + \frac{\sum_{t=1}^T \varepsilon_t}{2T} + O\left(\frac{1}{\sqrt{T}}\right).$$

Alternatively with the assumption that at each timestep $t$, $\hat{v}_{\boldsymbol{\alpha}_t}(\omega_t) \ge \rho$, $\bar{\lambda}$, satisfies

$$\max_{\boldsymbol{\alpha} \in \mathcal{K}} F(\boldsymbol{\alpha}, \bar{\lambda}) \le \rho + O\left(\frac{1}{\sqrt{T}}\right).$$

If Algorithm 2 is used on a $l_2$-SVM, the regret bound can be improved to be $O(\frac{\log T}{T})$.

*Proof.* We will make use of the regret bound of online gradient ascent (see, e.g., (Hazan, 2016)). Here we only prove the theorem in the case for $l_1$-SVM with box constraint $C$, under the assumption of $\varepsilon_t$-approximate optimality of Algorithm 1. Extending the proof to other cases is straightfoward.

**Lemma C.1** (Regret bound for online gradient ascent). *Let $D$ be the diameter of the constraint set $\mathcal{K}$, and $G$ a Lipschitz constant for an arbitrary sequence of concave functions $f_t(\boldsymbol{\alpha}) \stackrel{def}{=} F(\boldsymbol{\alpha}, \lambda_t)$ on $\mathcal{K}$. Online gradient ascent on $f_t(\boldsymbol{\alpha})$, with step size schedule $\eta_t = \frac{G}{D\sqrt{t}}$, guarantees the following for all $T \geq 1$:*

$$\frac{1}{T}\sum_{t=1}^{T} f_t(\boldsymbol{\alpha}_t) \geq \max_{\boldsymbol{\alpha}\in\mathcal{K}} \frac{1}{T}\sum_{t=1}^{T} f_t(\boldsymbol{\alpha}) - \frac{3GD}{2\sqrt{T}}.$$

Here, $D \leq C\sqrt{n}$ by the box constraint, and we have

$$G \leq \sup_{\substack{\boldsymbol{\alpha}\in\mathcal{K}, \\ \omega\in\mathbb{R}^d}} \|\mathbf{1} - 2\mathbf{Y}\operatorname{Re}(\langle \Phi_t, \mathbf{Y}\boldsymbol{\alpha}_t\rangle \overline{\Phi_t})\|_2$$
$$\leq (1+2C)\sqrt{n}.$$

Thus, our regret bound is $\frac{3n(C+2C^2)}{2\sqrt{T}}$; that is, for all $T \geq 1$,

$$\frac{1}{T}\sum_{t=1}^{T} F(\boldsymbol{\alpha}_t, \delta_{\omega_t} + \delta_{-\omega_t}) \geq \max_{\boldsymbol{\alpha}\in\mathcal{K}} \frac{1}{T}\sum_{t=1}^{T} F(\boldsymbol{\alpha}, \delta_{\omega_t} + \delta_{-\omega_t})) - \frac{3n(C+2C^2)}{2\sqrt{T}}.$$

Since at each timestep $t$, $\hat{v}_{\boldsymbol{\alpha}_t}(\omega_t) \geq \max_{\omega\in\mathcal{K}} \hat{v}_{\boldsymbol{\alpha}_t}(\omega) - \varepsilon_t$, we have by assumption

$$F(\boldsymbol{\alpha}_t, \delta_{\omega_t} + \delta_{-\omega_t}) \leq \min_{\omega\in\mathcal{K}} F(\boldsymbol{\alpha}_t, \delta_{\omega} + \delta_{-\omega}) + \varepsilon_t,$$

and by concavity of $F$ in $\boldsymbol{\alpha}$, we know that, for any fixed sequence $\boldsymbol{\alpha}_1, \boldsymbol{\alpha}_2, \ldots, \boldsymbol{\alpha}_T$,

$$\min_{\lambda\in\mathcal{L}} \max_{\boldsymbol{\alpha}\in\mathcal{K}} F(\boldsymbol{\alpha}, \lambda) \geq \min_{\lambda\in\mathcal{L}} \frac{1}{T}\sum_{t=1}^{T} F(\boldsymbol{\alpha}_t, \lambda),$$

it follows that

$$\min_{\lambda\in\mathcal{L}} \max_{\boldsymbol{\alpha}\in\mathcal{K}} F(\boldsymbol{\alpha}, \lambda) \geq \min_{\lambda\in\mathcal{L}} \frac{1}{T}\sum_{t=1}^{T} F(\boldsymbol{\alpha}_t, \lambda)$$
$$\geq \frac{1}{T}\sum_{t=1}^{T} \min_{\omega\in\mathcal{K}} F\left(\boldsymbol{\alpha}_t, \frac{\delta_\omega + \delta_{-\omega}}{2}\right)$$
$$\geq \frac{1}{2}\left(\frac{1}{T}\sum_{t=1}^{T} F(\boldsymbol{\alpha}_t, \delta_{\omega_t} + \delta_{-\omega_t}) - \frac{\sum_{t=1}^{T}\varepsilon_t}{T}\right)$$
$$\geq \max_{\boldsymbol{\alpha}\in\mathcal{K}} \frac{1}{2T}\sum_{t=1}^{T} F(\boldsymbol{\alpha}, \delta_{\omega_t} + \delta_{-\omega_t}) - \frac{3n(C+2C^2)}{4\sqrt{T}} - \frac{\sum_{t=1}^{T}\varepsilon_t}{2T}$$
$$= \max_{\boldsymbol{\alpha}\in\mathcal{K}} \frac{1}{2T}\sum_{t=1}^{T} F(\boldsymbol{\alpha}, \delta_{\omega_t} + \delta_{-\omega_t}) - \frac{3n(C+2C^2)}{4\sqrt{T}} - \frac{\sum_{t=1}^{T}\varepsilon_t}{2T}.$$

To complete the proof, note that for a given $\boldsymbol{\alpha} \in \mathcal{K}$, $\frac{1}{2T}\sum_{t=1}^{T} F(\boldsymbol{\alpha}, \delta_{\omega_t} + \delta_{-\omega_t}) = F(\boldsymbol{\alpha}, \bar{\lambda})$ by linearity of $F$ in the dual measure; here, $\bar{\lambda} = \frac{1}{2T}\sum_{t=1}^{T} \delta_{\omega_t} + \delta_{-\omega_t}$ is the approximate Nash equilibrium found by the no-regret learning dynamics.

$\square$

## D    PROOF OF THEOREM 4.3

In this section, we compute the Rademacher complexity of the composition of the learned kernel and the classifier, proving Theorem 4.3.

**Theorem 4.3.** For any SVM decision function $f : \mathcal{X} \to \mathbb{R}$ with a kernel $k_\lambda$ constrained by $\|\lambda\|_1 \leq R_\lambda$ trained on samples $S$ drawn i.i.d. from distribution $\mathcal{D}$, the generalization error is bounded by

$$\Pr_{(\mathbf{x},y)\sim\mathcal{D}}[yf(\mathbf{x}) \leq 0] \leq \min_\theta \frac{1}{n} \sum_{i=1}^n \mathbf{1}_{y_i f(x_i) \leq \theta} + \frac{6R_\omega R_x}{\theta} \sqrt{\frac{R_\lambda}{n}} + 3\sqrt{\frac{\log\frac{2}{\delta}}{n}}.$$

*Proof.* For a fixed sample $S = \{\mathbf{x}_i\}_{i=1}^n$, we compute the empirical Rademacher complexity of the composite hypothesis class. Below, $\sigma \in \mathbb{R}^n$ is a vector of i.i.d. Rademacher random variables.

$$\hat{\mathcal{R}}_S = \frac{1}{n}\mathbb{E}_\sigma\left[\sup_{\substack{\|\lambda\|_1\leq R_\lambda \\ \boldsymbol{\alpha}^\top G_{k_\lambda}\boldsymbol{\alpha}\leq 1}} \sum_{i=1}^n\sum_{j=1}^n \sigma_i\alpha_j k_\lambda(\mathbf{x}_i,\mathbf{x}_j)\right]$$

$$= \frac{1}{n}\mathbb{E}_\sigma\left[\sup_{\|\lambda\|_1\leq R_\lambda}\sup_{\boldsymbol{\alpha}^\top G_{k_\lambda}\boldsymbol{\alpha}\leq 1} \sigma^\top G_{k_\lambda}\boldsymbol{\alpha}\right]$$

$$= \frac{1}{n}\mathbb{E}_\sigma\left[\sup_{\|\lambda\|_1\leq R_\lambda}\sqrt{\sigma^\top G_{k_\lambda}\sigma}\right] = \frac{1}{n}\mathbb{E}_\sigma\left[\sqrt{\sup_{\|\lambda\|_1\leq R_\lambda}\sigma^\top G_{k_\lambda}\sigma}\right]$$

$$= \frac{1}{n}\mathbb{E}_\sigma\left[\sqrt{\sup_{\|\lambda\|_1\leq R_\lambda}\int_\Omega \lambda(\omega)\left|\sum_{i=1}^n \sigma_i e^{\iota\langle\omega,\mathbf{x}_i\rangle}\right|^2 d\omega}\right]$$

$$= \frac{1}{n}\mathbb{E}_\sigma\left[\sqrt{R_\lambda \sup_{\omega\in\Omega}\left|\sum_{i=1}^n \sigma_i e^{\iota\langle\omega,\mathbf{x}_i\rangle}\right|^2}\right]$$

$$= \frac{\sqrt{R_\lambda}}{n}\mathbb{E}_\sigma\left[\sup_{\omega\in\Omega}\sqrt{\left|\sum_{i=1}^n \sigma_i e^{\iota\langle\omega,\mathbf{x}_i\rangle}\right|^2}\right]$$

$$= \frac{\sqrt{R_\lambda}}{n}\mathbb{E}_\sigma\left[\sup_{\omega\in\Omega}\sqrt{\left|\sum_{i=1}^n \sigma_i \cos(\langle\omega,\mathbf{x}_i\rangle)\right|^2 + \left|\sum_{i=1}^n \sigma_i \sin(\langle\omega,\mathbf{x}_i\rangle)\right|^2}\right]$$

$$\leq \frac{\sqrt{R_\lambda}}{n}\mathbb{E}_\sigma\left[\sup_{\omega\in\Omega}\left(\left|\sum_{i=1}^n \sigma_i \cos(\langle\omega,\mathbf{x}_i\rangle)\right| + \left|\sum_{i=1}^n \sigma_i \sin(\langle\omega,\mathbf{x}_i\rangle)\right|\right)\right]$$

$$\leq \frac{\sqrt{R_\lambda}}{n}\mathbb{E}_\sigma\left[\sup_{\omega\in\Omega}\left|\sum_{i=1}^n \sigma_i \cos(\langle\omega,\mathbf{x}_i\rangle)\right|\right] + \frac{\sqrt{R_\lambda}}{n}\mathbb{E}_\sigma\left[\sup_{\omega\in\Omega}\left|\sum_{i=1}^n \sigma_i \sin(\langle\omega,\mathbf{x}_i\rangle)\right|\right].$$

Note that each term in the formula is the empirical Rademacher complexity of the composition of a 1-Lipschitz function (cosine or sine) with a linear function with $\ell_2$ norm bounded by $R_\omega$. By the composition property of Rademacher complexity, we have the following:

$$\frac{1}{n}\mathbb{E}_\sigma\left[\sup_{\omega\in\Omega}\left|\sum_{i=1}^n \sigma_i \cos(\langle\omega,\mathbf{x}_i\rangle)\right|\right]$$

$$\leq \frac{1}{n}\mathbb{E}_\sigma\left[\sup_{\omega\in\Omega}\sum_{i=1}^n \sigma_i \cos(\langle\omega,\mathbf{x}_i\rangle)\right] + \frac{1}{n}\mathbb{E}_\sigma\left[\sup_{\omega\in\Omega} -\sum_{i=1}^n \sigma_i \cos(\langle\omega,\mathbf{x}_i\rangle)\right]$$

$$\leq \frac{2R_\omega R_x}{\sqrt{n}},$$

and because $\sin(\cdot)$ is an odd function,

$$\frac{1}{n}\mathop{\mathbb{E}}_{\sigma}\left[\sup_{\omega\in\Omega}\left|\sum_{i=1}^{n}\sigma_i\sin(\langle\omega,\mathbf{x}_i\rangle)\right|\right]\leq\frac{R_\omega R_x}{\sqrt{n}}.$$

Putting everything together, we finally obtain that

$$\hat{\mathcal{R}}_S\leq 3R_\omega R_x\sqrt{\frac{R_\lambda}{n}},$$

from which we apply the result from Koltchinskii & Panchenko (2002) to conclude the theorem. $\quad\square$

## E  SAMPLE COMPLEXITY FOR THE FOURIER POTENTIAL

In this section, we prove the following theorem about concentration of the Fourier potential. It suffices to disregard the reweighting vector $\boldsymbol{\alpha}$; to recover a guarantee in this case, simply replace $\varepsilon$ with $\varepsilon/C^2$. Note that we only argue this in the translation-invariant case.

**Theorem 4.4.** Let $S=\{(\mathbf{x}_i,y_i)\}_{i=1}^{n}$ be a set of i.i.d. training samples. Then, if

$$n\geq\frac{8R_x^2 R_\omega^4(d\log\frac{2R_\omega}{\varepsilon}+\log\frac{1}{\delta})}{\varepsilon^2},$$

we have that with probability at least $1-\delta$, $\left|\frac{1}{n^2}v^{(n)}(\omega)-v_{\text{ideal}}(\omega)\right|\leq\varepsilon$ for all $\omega\in\Omega$ such that $\|\omega\|\leq R_\omega$.

Let $\mathbb{P}^{(n)},\mathbb{Q}^{(n)}$ denote the empirical measures from a sample $S$ of size $n$, arising from i.i.d. samples from the true distribution, whose classes have measures $\mathbb{P},\mathbb{Q}$, adopting the convention $\mathbb{P}(\mathcal{X})+\mathbb{Q}(\mathcal{X})=1$. Then, in expectation over the sampling, and adopting the same normalization conventions as in the paper, we have $\mathbb{E}[\mathbb{P}^{(n)}/n]=\mathbb{P}$ and $\mathbb{E}[\mathbb{Q}^{(n)}/n]=\mathbb{Q}$ for every $n$.

Let $v^{(n)}(\omega)$ denote the empirical Fourier potential, computed from $\mathbb{P}^{(n)},\mathbb{Q}^{(n)}$, so that we have $\mathbb{E}[v^{(n)}(\omega)/n^2]=v_{\text{ideal}}(\omega)$. The result follows from a concentration bound on $v^{(n)}(\omega)/n^2$. We first show that it is Lipschitz:

**Lemma E.1** (Lipschitzness of $v^{(n)}$ in $\omega$). *The function*

$$v^{(n)}(\omega)=\left|\sum_{i}^{n}y_i e^{\iota\langle\omega,\mathbf{x}_i\rangle}\right|^2 \tag{9}$$

*is $2n^2 R_x$-Lipschitz with respect to $\omega$.*

*Proof.* We have

$$\|\nabla_\omega v^{(n)}(\omega)\|=\left\|\nabla_\omega\left(\sum_{i,j\in[n]}y_i y_j e^{\iota\langle\omega,\mathbf{x}_i-\mathbf{x}_j\rangle}\right)\right\|$$

$$=\left\|\sum_{i,j\in[n]}y_i y_j i(\mathbf{x}_i-\mathbf{x}_j)e^{\iota\langle\omega,\mathbf{x}_i-\mathbf{x}_j\rangle}\right\|\leq n^2\cdot\max_{i,j}\|\mathbf{x}_i-\mathbf{x}_j\|\leq 2n^2 R_x.$$

$\square$

Thus, the Lipschitz constant of $v^{(n)}(\omega)/n^2$ scales linearly with the norm of the data, a safe assumption. Next, we show Lipschitzness with respect to a single data point:

**Lemma E.2** (Lipschitzness of $v^{(n)}$ in $\mathbf{x}_i$). $v^{(n)}(\omega)$ *is $2nR_\omega$-Lipschitz with respect to any $\mathbf{x}_i$.*

*Proof.* Similarly as above:

$$\|\nabla_{\mathbf{x}_i} v^{(n)}(\omega)\| = \left\| \nabla_{\mathbf{x}_i} \left( \sum_{i,j \in [n]} y_i y_j e^{\iota \langle \omega, \mathbf{x}_i - \mathbf{x}_j \rangle} \right) \right\|$$

$$\leq 2 \left\| \sum_{j \in [n]} y_i y_j\, i \omega e^{\iota \langle \omega, \mathbf{x}_i - \mathbf{x}_j \rangle} \right\| \leq 2n R_\omega.$$

$\square$

Now, we complete the proof. By Lemma E.2, replacing one $\mathbf{x}_i$ with another $\mathbf{x}'_i$ (whose norm is also bounded by $R_x$) changes $v^{(n)}(\omega)/n^2$ by at most $4 R_x R_\omega / n$. Then, by McDiarmid's inequality (McDiarmid, 1989), we have the following fact:

**Lemma E.3** (Pointwise concentration of $v^{(n)}$). *For all $\omega \in \Omega$ and $\varepsilon > 0$,*

$$\Pr \left[ \left| \frac{1}{n^2} v^{(n)}(\omega) - v_{\text{ideal}}(\omega) \right| \geq \varepsilon \right] \leq \exp \left( \frac{-n\varepsilon^2}{8 R_x^2 R_\omega^2} \right).$$

Now, let $W$ be an $(\varepsilon/R_\omega)$-net of the set $\{\omega \in \Omega : \|\omega\| \leq R_\omega\}$.

Applying the union bound on Lemma E.3 with $\omega$ ranging over each element of $W$, we have

$$\Pr \left[ \left| \frac{1}{n^2} v^{(n)}(\omega) - v_{\text{ideal}}(\omega) \right| \geq \varepsilon,\ \forall \omega \in R_\omega \right] \leq \Pr \left[ \left| \frac{1}{n^2} v^{(n)}(\omega) - v_{\text{ideal}}(\omega) \right| \geq \frac{\varepsilon}{R_\omega},\ \forall \omega \in W \right]$$

$$\leq |W| \cdot \exp \left( \frac{-n\varepsilon^2}{8 R_x^2 R_\omega^4} \right) \leq \left( \frac{2 R_\omega}{\varepsilon} \right)^d \exp \left( \frac{-n\varepsilon^2}{8 R_x^2 R_\omega^4} \right).$$

To make the LHS smaller than $\delta$, it suffices to choose

$$n \geq \frac{8 R_x^2 R_\omega^4 (d \log \frac{2 R_\omega}{\varepsilon} + \log \frac{1}{\delta})}{\varepsilon^2},$$

which is the claimed sample complexity. $\square$

