# OpenReview forum: "Not-So-Random Features"
_ICLR.cc/2018/Conference — Accept (Poster)_

### Official Review · AnonReviewer3 · 2017-11-26
**Interesting paper on kernel learning and random features**

**Rating:** 7
**Confidence:** 3

**Review:**

The paper proposes to learn a custom translation or rotation invariant kernel in the Fourier representation to maximize the margin of SVM. Instead of using Monte Carlo approximation as in the traditional random features literature, the main point of the paper is to learn these Fourier features in a min-max sense. This perspective leads to some interesting theoretical results and some new interpretation. Synthetic and some simple real-world experiments demonstrate the effectiveness of the algorithm compared to random features given the fix number of bases.

I like the idea of trying to formulate the feature learning problem as a two-player min-max game and its connection to boosting. As for the related work, it seems the authors have missed some very relevant pieces of work in learning these Fourier features through gradient descent [1, 2]. It would be interesting to compare these algorithms as well.

[1] Zichao Yang, Marcin Moczulski, Misha Denil, Nando de Freitas, Alex Smola, Le Song, Ziyu Wang. Deep Fried Convnets. ICCV 2015.
[2] Zichao Yang, Alexander J. Smola, Le Song, Andrew Gordon Wilson. A la Carte — Learning Fast Kernels. AISTATS 2015.

---

> ### Author Response · Authors · 2017-12-18
> **Response to AnonReviewer3**
>
> Similar to our work, paper [2] also considers learning the Fourier spectrum of a shift-invariant kernel, where the spectrum is parameterized as a mixture of (a fixed number of) Gaussians or a piecewise linear function, which can definitely fit in our min-max formulation. However, in comparison, our end-to-end method is more general since it doesn’t rely on any specific parameterization. Paper [1] is also interesting and relevant since it draws connections between spectrally learned kernel machines and deep neural networks. As we mention in the conclusion, it is an exciting future direction to build our method into a deep neural network. We thank the reviewer for pointing out these references, and we’ll add them to the related work section.

---

### Official Review · AnonReviewer1 · 2017-11-28
**interesting work but need more comparison**

**Rating:** 6
**Confidence:** 5

**Review:**


In this paper, the authors proposed an interesting algorithm for learning the l1-SVM and the Fourier represented kernel together. The model extends kernel alignment with random feature dual representation and incorporates it into l1-SVM optimization problem. They proposed algorithms based on online learning in which the Langevin dynamics is utilized to handle the nonconvexity. Under some conditions about the quality of the solution to the nonconvex optimization, they provide the convergence and the sample complexity. Empirically, they show the performances are better than random feature and the LKRF.

I like the way they handle the nonconvexity component of the model. However, there are several issues need to be addressed.

1, In Eq. (6), although due to the convex-concave either min-max or max-min are equivalent, such claim should be explained explicitly.

2, In the paper, there is an assumption about the peak of random feature "it is a natural assumption on realistic data that the largest peaks are close to the origin". I was wondering where this assumption is used? Could you please provide more justification for such assumption?

3, Although the proof of the algorithm relies on the online learning regret bound, the algorithm itself requires visit all the data in each update, and thus, it is not suitable for online learning. Please clarify this in the paper explicitly.

4, The experiment is weak. The algorithm is closely related to boosting and MKL, while there is no such comparison. Meanwhile, Since the proposed algorithm requires extra optimization w.r.t. random feature, it is more convincing to include the empirical runtime comparison.

Suggestion: it will be better if the author discusses some other model besides l1-SVM with such kernel learning.

---

> ### Author Response · Authors · 2017-12-18
> **Response to AnonReviewer1**
>
>
> @1: We mention the minimax theorem in the proof, in the appendix. We can add a brief clarification in the main paper.
>
> @2: The only assumption (for the theorems to hold) is that Algorithm 1 finds an eps-approximate global maximum. Our discussion on band-limitedness of real-world data is simply to argue that this non-convex problem is plausibly easy on realistic optimization landscapes: low-frequency features (on the same scale as the RBF bandwidth parameter; see Appendix A.1) are informative.
>
> @3: That’s correct, just as in boosting. We are happy to find a way to further emphasize the distinction, to reduce confusion.
>
> @4:
> - As mentioned in the paper, MKL methods take >100 times longer on datasets as large as CIFAR-10.
> - Unlike the selling point of methods such as LKRF and Quasi-Monte Carlo, our method has much greater expressivity (thus evidently saturates at a much higher accuracy). Hence, the value of a quantitative wall clock time comparison is unclear. We believe that the existing discussion (primal like RF; parallelizable; reasonable wall clock time in practice) suffices to address qualitative questions on efficiency as compared to other paradigms.
> - Is there a specific boosting method the reviewer believes to be related enough, so as to require an end-to-end comparison? We found that it’s unclear how to choose an ensemble in boosting for fair comparison with learning an optimal translation-invariant kernel (an infinite-dimensional continuous family). As far as we know, though our theoretical analysis bears a strong relationship to boosting, the end-to-end methodology is somewhat dissimilar.
>
> @Suggestion: We agree that considering state-of-the-art settings and applications is an important and interesting direction (as we note in the conclusion). As we mentioned in another review, any convex kernel machine admitting a dual could fit in our min-max formulation, while the min-max SVM objective captures the structure of learning a kernel for any such kernel machine.

---

### Official Review · AnonReviewer2 · 2017-12-12
**Solid submission but the authors overclaim their contribution**

**Rating:** 4
**Confidence:** 5

**Review:**

In this paper the authors consider learning directly Fourier representations of shift/translation invariant kernels for machine learning applications. They choose the alignment of the kernel to data as the objective function to optimize. They empirically verify that the features they learned lead to good quality SVM classifiers. My problem with that paper is that even though at first glance learning adaptive feature maps seems to be an attractive approach, authors' contribution is actually very little. Below I list some of the key problems. First of all the authors claim in the introduction that their algorithm is very fast and with provable theoretical guarantees. But in fact later they admit that the problem of optimizing the alignment is a non-convex problem and the authors end up with a couple of heuristics to deal with it. They do not really provide any substantial theoretical justification why these heuristics work in practice even though they observe it empirically. The assumptions that large Fourier peaks happen close to origin is probably well-justified from the empirical point of view, but it is a hack, not a well established well-grounded theoretical method (the authors claim that in their experiments they found it easy to find informative peaks, even in hundreds of dimensions, but these experiments are limited to the SVM setting, I have no idea how these empirical findings would translate to other kernelized algorithms using these adaptive features).  The Langevin dynamics algorithm used by the authors to find the peaks (where the gradient is available) gives only weak theoretical guarantees (as the authors actually admit) and this is a well known method, certainly not a novelty of that paper. Finally, the authors notice that "In the rotation-invariant case, where Ω is a discrete set, heuristics are available". That is really not very informative (the authors refer to the Appendix so I carefully read that part of the Appendix, but it is extremely vague, it is not clear at all how the Langevin dynamics can be "emulated" by a discrete Markov chain that mixes fast; the authors do not provide any justification of that approach, what is the mixing time ?; how the "good emulation property" is exactly measured ?).  In the conclusions the authors admit that: "Many theoretical questions remain, such as accelerating the search for Fourier peaks". I think that the problem of accelerating this approach is a critical point that this publication is missing. Without this, it is actually really hard to talk about general mechanism of learning adaptive Fourier features for kernel algorithms (which is how the authors present their contribution); instead we have a method heavily customized and well-tailored to the (not particularly exciting) SVM scenario (with optimization performed by the standard annealing method; it is not clear at all whether for other downstream kernel applications this approach for optimizing the alignment would provide good quality models) that uses lots of task specific hacks and heuristics to efficiently optimize the alignment. Another problem is that it is not clear at all to me how authors' approach can be extended to non shift-invariant kernels that do not benefit from Bochner's Theorem. Such kernels are very related to neural networks (for instance PNG kernels with linear rectifier nonlinearities correspond to random layers in NNs with ReLU) and in the NN context are much more interesting that radial basis function or in general shift-invariant kernels. A general kernel method should address this issue (the authors just claim in the conclusions that it would be interesting to explore the NN context in more detail).

To sum it up, it is a solid submission, but in my opinion without a substantial contribution and working only in a  very limited setting when it is heavily relying on many unproven hacks and heuristics.

---

> ### Author Response · Authors · 2017-12-18
> **Response to AnonReviewer2**
>
> The reviewer’s summary (“they choose the alignment of the kernel to data as the objective function to optimize”) appears to have missed our main contribution, which is our formulation of the kernel learning problem as a min-max game whose Nash equilibrium gives the optimal kernel. This is harder than simply maximizing kernel alignment, which is the objective considered by most previous works; it is also more useful and principled, as this optimizes the generalization bound directly. Our contribution lies in the provable black-box reduction from computing this Nash to solving *adversarially weighted* instances of kernel alignment. We are concerned that this confusion possibly underlies the reviewer’s score and conclusion.
>
>
> @Langevin: We never claim that the use of Langevin gradient is the crux (or novelty) of the paper. Again, our contribution is giving a theoretically sound reason and strong empirical evidence to use multiple rounds of *adversarially weighted* kernel alignment rather than the uniformly weighted one (which the reviewer possibly had in mind). The role of Langevin is to provide an end-to-end pipeline for this reduction which works well in practice; we don’t believe that this makes the entire methodology a “hack”.
>
> @Spherical harmonics: The mention of the Markov chain on spherical harmonic indices should be viewed as a side note for practical implementation. Finding a band-limited Fourier peak in discrete space is *easier* in practice, even though there is no gradient; we first mention that enumeration of indices is possible, with no possible optimization error (unlike the continuous case). By “emulating” Langevin dynamics, we refer to a random walk on the lattice of valid indices. Again, the role of all discussion here on Monte Carlo for optimization is to complete a practical end-to-end pipeline, in this case for the rotation-invariant version.
>
> @Non-convexity: An end-to-end polynomial-time guarantee with no assumptions would be a *significant* breakthrough: it would entail an optimal way to train two-layer neural nets with cosine activations. Our reduction connects this daunting task to a classic and natural non-convex problem (high-dimensional FFT), an active area of theoretical research as pointed out in the conclusion.
>
> @Fourier assumption: We don’t believe that the hypothesis that large Fourier peaks are close to the origin is such a controversial one. This is essentially the same as why \ell_2 regularization is widely adopted and well-justified (see paper [1]). Algorithmically, as we point out in a response to another review, the assumption does not even need to show up. However, we could allay this concern more rigorously via an explicitly-enforced \ell_2 regularizer or constraint. Since multiple reviewers have mentioned this, we’ll revise the manuscript to clarify.
>
> @Downstream kernel methods: Our method applies to downstream supervised kernel algorithms at least as well as kernel alignment, a widely considered objective in MKL. (Note that kernel alignment is a special case of our method, with T=1 round of boosting.) We found that a comprehensive presentation and evaluation of the plethora of kernel methods would distract from the main focus. We further note that the min-max formulation is possible for any convex kernel machine admitting a dual: e.g. support vector regression, kernel ridge regression, and statistical testing (which we can add to the appendix). The min-max SVM objective captures the structure of learning a kernel for any such kernel machine.
>
> @More expressive kernels: Shift-invariant kernels are very expressive and general, compared to kernel families with any comparable theory (existing work in random features). In ICML/NIPS/JMLR 2016-2017, there is a huge amount of research solely on efficiently approximating (not learning) a *single* RBF kernel (see, e.g. papers [3, 4, 5, 6]). We also agree that non-shift-invariant kernels are a very interesting direction to consider. However, much care must be taken, as some restriction on the kernel must be chosen; otherwise, overfitting is inevitable (due to the no-free-lunch theorem).
>
> @ReLU-NN: To fully address the issue of learning a two-layer ReLU neural network is a well-known NP-hard problem (paper [2]). We agree that hardness results shouldn’t prevent us from advancing. However, any non-trivial improvement on this problem should surely deserve a separate paper, which is why we mention it as an interesting future direction.
>
>
> [1] Kakade et al. On the Complexity of Linear Prediction: Risk Bounds, Margin Bounds, and Regularization, NIPS 2008.
> [2] Klivans, Sherstov. Cryptographic Hardness for Learning Intersections of Halfspaces, FOCS 2006.
> [3] Yu et al. Orthogonal random features, NIPS 2016.
> [4] Lyu. Spherical Structured Feature Maps for Kernel Approximation, ICML 2017.
> [5] Avron et al. Quasi-Monte Carlo Feature Maps for Shift-Invariant Kernels, JMLR 2016.
> [6] Dao et al. Gaussian Quadrature for Kernel Features, NIPS 2017.

---

### Public Comment · (anonymous) · 2017-12-31
**Pretty interesting idea.. But why Langevin dynamics?**

The two player game formulation for SVM is pretty cool! Computing the optimal feature by searching the largest Fourier peak under adversarially picked alpha also looks interesting.

However, it is not clear to me why the authors are using Langevin dynamics for searching the peak -- why not SGD? Is the additional noise in Langevin dynamics more useful empirically?

---

> ### Author Response · Authors · 2018-01-04
> **Response**
>
> Thanks for your interest!
>
> Our algorithm for maximizing the Fourier potential picks the highest peak along the trajectory of Langevin dynamics (rather than returning the final $\omega$). This seems necessary for refining the margin effectively, and requires evaluation of the full Fourier potential at each iteration. Evaluating the Fourier potential function requires a full pass over the data (rather than an SGD minibatch).
>
> Furthermore, the Online Gradient Ascent steps need to be taken on all entries of $\alpha$, which also requires a full pass over data; subsampled analogues such as Online Coordinate Ascent give a worse regret bound.
>
> Since each iteration requires a full pass over data anyway, Langevin dynamics is preferred over SGD.

---

### Author Response · Authors · 2018-01-05
**Revised manuscript**

Thanks for the reviews and comments!

We've made a few minor revisions to the manuscript, mostly for clarity and brevity.

---

### Decision · Program_Chairs · 2018-01-29
**ICLR 2018 Conference Acceptance Decision**

**Decision:**

Accept (Poster)

**Comment:**

New effective kernel learning methods are very well aligned with ICLR's focus on Representation Learning. As a reviewer pointed out, not all aspects of the paper are algorithmically "clean". However, the proposed approach is natural and appears to give consistent improvements over a couple of expected baselines. The paper could be strengthened with more comparisons against other kernel learning methods, but acceptance at ICLR-2018 will increase the diversity of the conversation around advances in Representation Learning.